# A phosphorylation switch turns a positive regulator of phototropism into an inhibitor of the process

Paolo Schumacher[1], Emilie Demarsy [1,3], Patrice Waridel [2], Laure Allenbach Petrolati [1], Martine Trevisan [1] & Christian Fankhauser [1]

Phototropins are light-activated protein kinases, which contribute to photosynthesis optimization both through enhancement of photon absorption when light is limiting and avoidance responses in high light. This duality is in part endowed by the presence of phototropins with different photosensitivity (phot1 and phot2). Here we show that phot1, which senses low light to promote positive phototropism (growth towards the light), also limits the response in high light. This response depends in part on phot1-mediated phosphorylation of Phytochrome Kinase Substrate 4 (PKS4). This light-regulated phosphorylation switch changes PKS4 from a phototropism enhancer in low light to a factor limiting the process in high light. In such conditions phot1 and PKS4 phosphorylation prevent phototropic responses to shallow light gradients and limit phototropism in a natural high light environment. Hence, by modifying PKS4 activity in high light the phot1-PKS4 regulon enables appropriate physiological adaptations over a range of light intensities.

[1] Center for Integrative Genomics, Faculty of Biology and Medicine, University of Lausanne, Genopode Building, 1015 Lausanne, Switzerland. [2] Protein Analysis Facility, Center for Integrative Genomics, Faculty of Biology and Medicine, University of Lausanne, 1015 Lausanne, Switzerland. [3]Present address: Department of Botany and Plant Biology, University of Geneva, 1211 Geneva 4, Switzerland. Correspondence and requests for materials should be addressed to C.F. (email: christian.fankhauser@unil.ch)

Plants transform solar energy into chemical energy in the process of photosynthesis. In order to optimize this process several classes of photosensory receptors monitor the light environment to modulate growth and development accordingly[1–3]. Blue light photoreceptors from the phototropin family optimize photosynthesis through several mechanisms[4,5]. Angiosperms typically possess two phototropins: phot1 working over a broad range of fluence rates and phot2 that is rather specialized in high light responses[6]. This pair of phototropins controls physiological responses such as phototropism (growth oriented by the light direction), stomatal opening, chloroplast positioning, and leaf flattening with the importance of phot1 and phot2 depending on the light intensity and/or physiological response[6].

Phot1 and phot2 differentially contribute to the phototropic response. Phot1 controls phototropism towards low fluence rates while at higher light intensity both phototropins mediate the response[7]. This physiological adaptation allows optimal organ positioning through directional growth to maximize photon capture by leaves when light is limiting[8]. The phototropic response is developmentally regulated with green plantlets relying on partially different mechanisms than etiolated seedlings[9,10]. A possible explanation for different response patterns and mechanisms is that etiolated seedlings need to rapidly establish photo-autotrophy and are therefore very sensitive to light[8,10,11]. In contrast, in green seedlings light gradients do not always trigger a strong phototropic response in particular in favorable light conditions[2,3,10]. How this is controlled and whether it directly involves the phototropins remains poorly understood.

Phototropins are composed of a pair of amino-terminal photosensory Light-Oxygen-Voltage (LOV1 and LOV2) domains and a carboxyl terminal AGC protein kinase domain[4]. Blue light induces the protein kinase activity through structural rearrangements involving two α helices flanking the LOV2 domain[4,12]. In Arabidopsis the protein kinase activity of both phototropins is essential for all tested physiological responses[4,13,14]. Light triggers phototropin phosphorylation at multiple residues including serines in an activation loop leading to enhanced kinase and physiological activity[13,14]. Phot1 acts at the plasma membrane where it interacts with early signaling components from two protein families: NPH3/RPT2-like (NRL) and Phytochrome Kinase Substrate (PKS1, 2, and 4)[15–20]. Following phot1 photo-excitation NPH3 is rapidly dephosphorylated, an event associated with NPH3 release from the plasma membrane and phot1 signaling desensitization[21,22]. A limited number of phot1 substrates are known[4]. We previously showed that PKS4 is part of this small group; however, the functional significance of this signaling event remains unclear[4,19]. The auxin transporter ABCB19 and the protein kinase BLUS1 are other known substrates of phot1[9,23,24]. While phosphorylation of ABCB19 appears to inhibit its activity, BLUS1 phosphorylation is essential for its activation[9,23]. Hence, as observed for other protein kinases, the consequences of phot1-mediated phosphorylation depend on the substrate.

In order to understand the functional role of phot1-mediated PKS4 phosphorylation, we identify a PKS4 serine (S) to alanine (A) mutant, which prevents light-induced PKS4 phosphorylation. Our studies indicate that PKS4 phosphorylation limits the phototropic response in strong light. This response depends on phot1, thereby revealing an unexpected role for phot1 in limiting phototropism. Hence phot1 and PKS4, two elements that are essential to respond to low light, have a dual role and also limit the response to high light.

## Results

### S299 is essential for light-induced PKS4 phosphorylation. PKS4 exists in two isoforms that can be distinguished by their mobility

on SDS-PAGE gels: the faster migrating PKS4D (the only form present in the dark) and PKS4L a light-induced phot1-mediated slower migrating phosphorylated isoform[19]. To study the role of PKS4L we searched for PKS4 phosphorylation sites. Initially, we focused on nine residues because they were either found in previous large-scale phospho-proteomics studies or because they are conserved putative phot1/AGC kinase phosphorylation sites[25–28] (Fig. 1a). We generated an "All-A" mutant in which S80, S85, S86, S122, S129, S130, T258, S259, and S299 were mutated into A. This "All-A" mutant and a wild-type (WT) PKS4 were epitope tagged with a triple HA tag at the carboxy-terminus and expressed under the control of a 1.5 kb PKS4 promoter. Both constructs were introduced into a pks4 mutant to study the light-induced appearance of PKS4L. Etiolated seedlings were treated with blue light to trigger phot1-dependent PKS4 phosphorylation, which was followed using western blotting[19]. In contrast to its wild-type counterpart, we did not detect the PKS4L isoform at any time point following illumination in the "All-A" mutant (Supplementary Fig. 1A). This experiment showed that this "All-A" mutant comprised at least one residue essential for the appearance of the slower migrating phosphorylated PKS4L isoform.

To identify the residue(s) undergoing light-regulated phosphorylation, we used immunoprecipitation followed by mass spectrometry and concentrated our analysis on the nine residues mutated in the "All-A" variant. We used etiolated seedlings without or with a short strong blue light pulse leading to the appearance of PKS4L[19]. This analysis showed that the peptide comprising S299 was detected in its unphosphorylated form both in dark-grown and light-treated seedlings while the phosphorylated peptide was only present in light-treated seedlings (Fig. 1b). To test whether S299 is required for the formation of PKS4L, we made the following transgenic lines: PKS4-S299A and PKS4-S299D with a C-terminal triple HA tag expressed from the PKS4 promoter transformed into pks4. We refer to these lines as S299A and S299D respectively. We found that plants expressing S299A failed to accumulate PKS4L in the light (Fig. 1c). Western blot analysis revealed that in the dark S299D migrated more slowly than the wild-type protein. Moreover, S299D migration was not light regulated and its apparent molecular weight was similar to PKS4L in the "WT" transgenic line (Fig. 1d). Taken together, these results show that S299 is phosphorylated in a light-dependent fashion and is essential for the appearance of the light-induced phosphorylated PKS4L isoform. Moreover, mutating this residue into aspartate led to the constitutive presence of a PKS4L-like isoform. We therefore focused our functional analysis on plants expressing S299 variants.

To test whether the substitution of S299 alter known features of PKS4 we analyzed subcellular localization and protein−protein interactions[19]. Association of PKS4 with the plasma membrane was determined in plants expressing PKS4-GFP variants by confocal laser-scanning microscopy and biochemically. PKS4-GFP localization in hypocotyl cells was not affected by blue light treatments or S299 mutations (Supplementary Fig. 1B). Consistent with microscopic data, biochemical fractionation showed strong PKS4-GFP enrichment in microsomal fractions in etiolated and light-treated samples of all genotypes (Supplementary Fig. 1C). To assess PKS4 interaction with phot1 and NPH3[19] we used co-immunoprecipitation and found that PKS4-S299A and PKS4-S299D interacted with phot1 and NPH3 as the wild type (Supplementary Fig. 1D). These experiments suggest that these mutations do not lead to substantial structural modification of PKS4.

To assess the function of PKS4 variants we first determined the phosphorylation state of S299D, which migrates like PKS4L irrespective of the light treatment (Fig. 1d). This was analyzed on SDS-PAGE gels following phosphatase treatments to detect

mobility shifts associated with phosphorylation. This experiment showed that in contrast to its WT counterpart, S299D no longer displayed light-regulated phosphorylation and that S299D

**a**

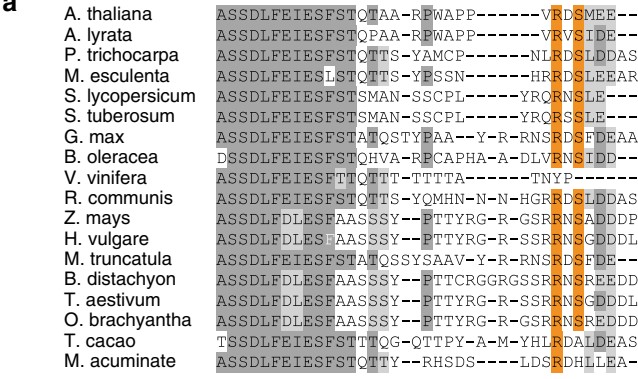

**b**

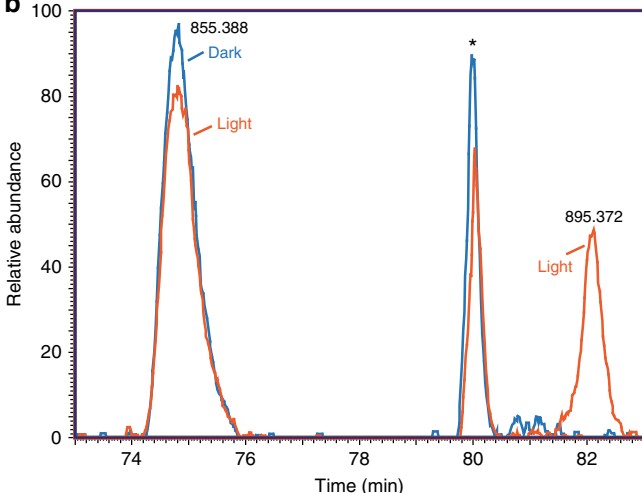

**c**

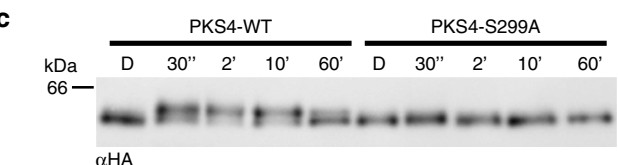

**d**

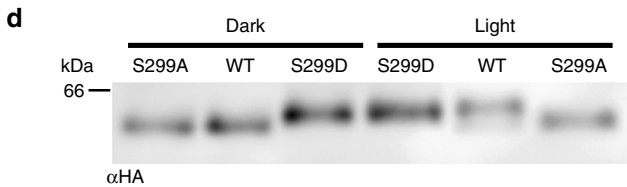

**Fig. 1** S299 is required for light-induced PKS4 phosphorylation. **a** Alignment of PKS4 orthologs from angiosperm. The S299 PKA consensus sequence is highlighted in orange. Identical residues are marked in dark gray, light gray indicates similar residues. **b** Mass spectrometry analysis of the unphosphorylated (left, $m/z$: 855.388) and phosphorylated peptide (right, $m/z$: 895.372) APPVRDSMEETVSEY containing S299 from extracts of dark-grown seedlings (in blue) and light-grown seedlings (in orange). The peak in the middle (marked with an asterisk) is from a different peptide. **c** Western blot probed with anti-HA antibodies from *pks4* PKS4-WT and *pks4* S299A samples of etiolated seedlings treated with 15 μmol m$^{-2}$ s$^{-1}$ blue light for 0, 2, 10, or 60 min. PKS4D and PKS4L mark the slower and faster migrating isoforms respectively. **d** Western blot probed as in (**c**) from *pks4* PKS4-WT, *pks4* S299A, and *pks4* S299D samples of etiolated seedlings treated with 150 μmol m$^{-2}$ s$^{-1}$ blue light for 30 s or kept in darkness

migrated more slowly than the WT in its unphosphorylated form (Supplementary Fig. 1E). We then analyzed NPH3 dephosphorylation, which is a rapid phot1-mediated response[21]. NPH3 dephosphorylation occurred in all genotypes including *pks4* indicating that this response is independent of PKS4 (Supplementary Fig. 1E, F). The function of PKS4 variants was determined by assessing their ability to rescue the low-light-specific phototropic defect of *pks4*[29]. We found that the mutated versions of PKS4 complemented the *pks4* mutant as well as the wild type (Supplementary Fig. 2A). Multiple independent insertion lines were analyzed and when compared to protein expression levels, we observed that the phototropic response was relatively insensitive to PKS4 protein levels (Supplementary Fig. 2B). Collectively our data show that S299A and S299D have a prominent effect on PKS4D and PKS4L formation. However, these mutations have no effect on tested PKS4 features and NPH3 dephosphorylation, an early phot1-mediated response (Supplementary Fig. 1). Finally, we conclude that the formation of PKS4L is not required for PKS4 to promote phototropism in low light (Supplementary Fig. 2A).

**PKS4L negatively regulates phototropism**. To reveal a potential role of PKS4L, we concentrated on higher light responses as PKS4L accumulates to higher levels with increasing light intensity[19], exemplified by the large difference in PKS4L formation between etiolated seedlings treated with 0.1 and 10 μmol m$^{-2}$ s$^{-1}$ blue light (Supplementary Fig. 2C). We compared the kinetics of phototropism at those fluence rates and observed that a line expressing WT PKS4 (WT-2) had a slower phototropic response at 10 than at 0.1 μmol m$^{-2}$ s$^{-1}$ (Fig. 2a). Interestingly, the PKS4-S299A mutant, which no longer accumulated PKS4L, exhibited similar bending kinetics in both conditions and resembled the phenotype of WT plants at the low fluence rate (Fig. 2b, S2D, Supplementary Fig. 2E). This suggests that the presence of PKS4L in WT plants slows down phototropism. Consistent with this hypothesis, the phosphomimic line PKS4-S299D showed a slower phototropic response at 0.1 μmol m$^{-2}$ s$^{-1}$ compared to the WT (Supplementary Fig. 2F). Consistent with previous observation we confirmed that PKS4 is required for the fast phototropic response in response to a low fluence rate (Supplementary Fig. 2G)[29]. To further test the notion that PKS4L inhibits phototropism, we analyzed phototropism triggered by light pulses. In response to a low light fluence phototropism is proportional to the amount of light, a phase known as first-degree positive curvature[30]. Beyond a certain fluence phototropism is inhibited, known as the refractory phase, suggesting the existence of an inhibitory process[30]. To determine whether PKS4 phosphorylation contributes to this regulation etiolated seedlings were irradiated with light pulses of either 0.03 μmol m$^{-2}$ s$^{-1}$ (to trigger first positive curvature) or 3 μmol m$^{-2}$ s$^{-1}$ (enough to trigger the refractory phase). Col-0, PKS4-WT, and PKS4-S299A behaved the same at 0.03 μmol m$^{-2}$ s$^{-1}$. At 3 μmol m$^{-2}$ s$^{-1}$, bending was inhibited in Col-0 and PKS4-WT. In contrast, phototropic bending was not significantly inhibited in the S299A line, supporting an inhibitory role of PKS4L during phototropism (Fig. 2c). Consistent with previous observations, *pks4* showed reduced bending at 0.03 and no significant inhibition of the response at 3 μmol m$^{-2}$ s$^{-1}$[19]. We previously showed that the type 2A protein phosphatase inhibitor cantharidin inhibits PKS4L dephosphorylation and phototropism[19]. By comparing the effect of cantharidin in plants expressing WT and S299A, we observed that in our assay condition the effect of cantharidin depended on the presence of PKS4L (Supplementary Fig. 2H). Taken together, our results show that in etiolated seedlings the accumulation of PKS4L acts as a negative regulator of phototropism (Fig. 2 and Supplementary Fig. 2).

**phot1-mediated PKS4L accumulation inhibits phototropism.**
To further test the role of PKS4L we decided to determine whether at high light intensities typical of an outdoor environment the formation of PKS4L also modulates phototropism. We measured the light spectrum on a sunny day in Lausanne and found that blue light (400–500 nm) was ~660 $\mu$mol m$^{-2}$ s$^{-1}$ (PAR was ~2000 $\mu$mol m$^{-2}$ s$^{-1}$). To analyze phototropism in response to high light we used green seedlings because in nature it is unlikely that etiolated seedlings suddenly get exposed to such strong unilateral light. However, depending on their position

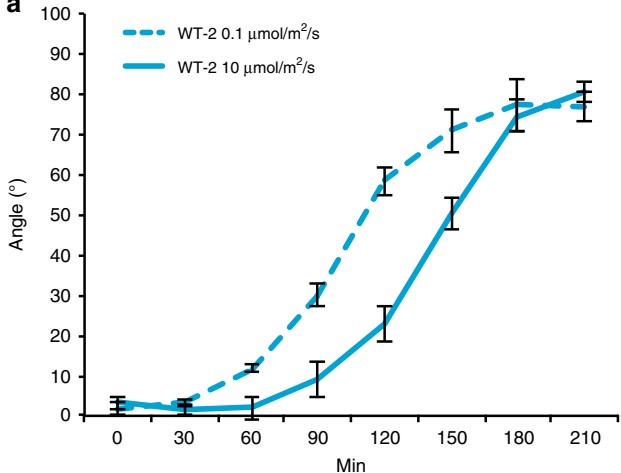

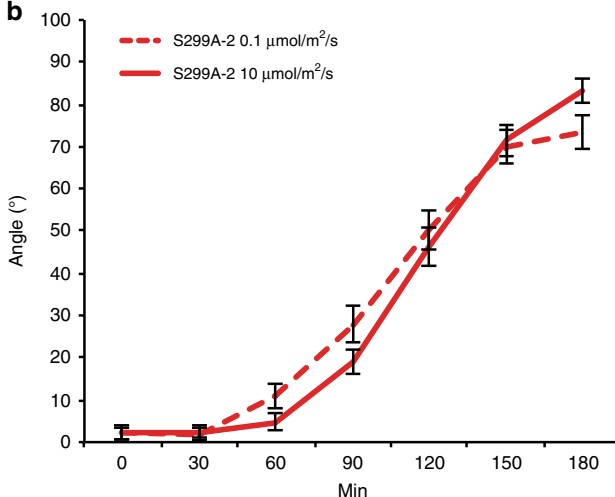

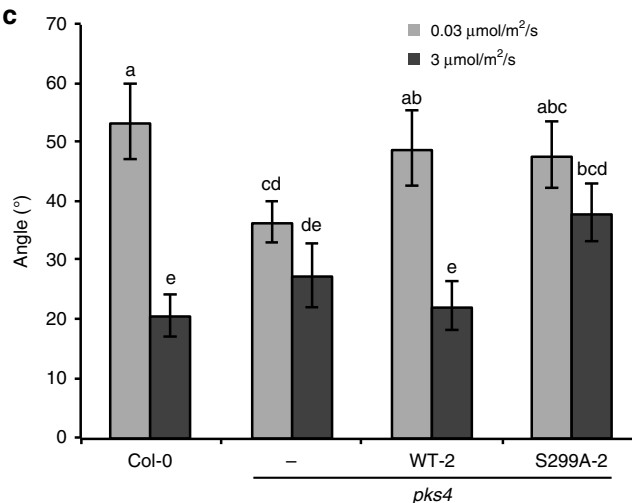

within the plant community seedlings may experience periods of the day with strong light gradients[3]. Interestingly, the phototropic response of green seedlings significantly declined at higher fluence rates (Fig. 3a). A reduction of phototropism was similarly observed in *pks4* mutant and *pks4*-expressing PKS4-WT (Fig. 3b, Supplementary Fig. 3A). However, this inhibition was not observed in S299A lines, but was also strong in the S299D expressing mutant (Fig. 3b, Supplementary Fig. 3B). To further test the hypothesis that PKS4-S299D inhibits phototropism we crossed the PKS4-S299D and the PKS4-S299A lines with *phot1* to analyze phototropism in a sensitized genetic background. This experiment showed that *phot1pks4* expressing PKS4-S299D had a strongly reduced response while *phot1pks4* PKS4-S299A responded as *phot1* (Supplementary Fig. 3C). These experiments are consistent with PKS4L contributing to phototropism inhibition in high light. Moreover, the *pks4* phenotype indicates that other processes also modulate this response. Finally, our data indicate that the S299D mutant maintains some level of light regulation of phototropism despite the lack of an obvious mobility shift of S299D on SDS-PAGE gels (Figs. 1 and 3b, Supplementary Fig. 3B).

We previously showed that in etiolated seedlings phot1 phosphorylates PKS4[19]. To determine which phototropin(s) control PKS4 phosphorylation in green seedlings we analyzed PKS4 migration on western blots in de-etiolated plantlets treated with strong blue light. We found that PKS4L was present in Col-0 and *phot2* but not in *phot1* and the *phot1phot2* double mutants showing that the presence of PKS4L also depends on phot1 in green seedlings (Fig. 3c). The phenotype of S299A indicates that PKS4L inhibits phototropism suggesting that in *phot1* strong light may no longer inhibit phototropism. Interestingly, phototropism was reduced at the higher fluence rate both in Col-0 and *phot2* while in *phot1*, phototropism was wild type at the lower fluence rate but not inhibited at 600 $\mu$mol m$^{-2}$ s$^{-1}$ (Fig. 3d). As reported previously the phototropic response of *phot1phot2* double mutant was strongly impaired[7,10] (Fig. 3d). To test the relevance of this data in more natural conditions we performed phototropism experiments in the greenhouse (Supplementary Fig. 3D). On the day of the experiment shown in Fig. 3e we recorded ~250 $\mu$mol m$^{-2}$ s$^{-1}$ blue light. Consistent with the inhibitory role of PKS4L, PKS4-S299A and *phot1* plants had a stronger phototropic response than Col-0 and *phot2* (Fig. 3e). In a separate experiment we compared phototropism in the greenhouse of Col-0, *pks4* and *pks4* transformed either with WT, S299A or S299D. Blue light levels were a bit lower that day (~200 $\mu$mol m$^{-2}$ s$^{-1}$); however, the S299A had the strongest phototropic response while phototropism was the weakest in S299D (Supplementary Fig. 3D, experimental design shown in Supplementary Fig. 3E). Collectively our data are consistent with

**Fig. 2** The phototropic response of S299A mutants is less sensitive to increased fluences. **a**, **b** Phototropism time course of 3-day-old etiolated seedlings of *pks4* WT-2 and *pks4* S299-A2 treated with 0.1 or 10 $\mu$mol m$^{-2}$ s$^{-1}$ lateral blue light for the indicated time. $n = 15–29$, data represent means, error bars correspond to 2× standard error. **c** Phototropic curvature of 3-day-old etiolated seedlings of Col-0, *pks4*, *pks4* WT-2, and *pks4* S299-A2. Seedlings were exposed to two 10 min pulses of unilateral blue light at either 0.03 or 3 $\mu$mol m$^{-2}$ s$^{-1}$, separated by 50 min of darkness. One hour prior to the first blue light pulse plants were treated for 1 min with 30 $\mu$mol m$^{-2}$ s$^{-1}$ red light to enhance the phototropic response by activating phytochromes. After the second pulse plants were kept in darkness for 3 h prior to measurement of growth re-orientation. $n = 15–20$, data represent means, error bars correspond to 2× standard error, means with same letter are not significantly different ($p > 0.05$, two-way ANOVA test)

a model where in response to high light phot1 inhibits phototropism through phosphorylation of PKS4.

We reasoned that limiting phototropism in high light might be important to properly respond to light gradients, which in nature are less extreme than those obtained using a unidirectional light source[3]. We therefore treated green seedlings with two blue light sources emitting different light intensities from opposite sides taking advantage from the fact that the fluence rate diminishes as one moves away from the light source (Fig. 4a). Depending on the position of the seedling on the plate, it was either exposed to similar levels coming from each side (position R) up to a 2/1 ratio (position L, Fig. 4a). Seedlings were treated with different total blue light intensities while maintaining the same gradient across the plate. In Col-0 and pks4 complemented by the wild-type PKS4 high overall light levels inhibited the response to the light gradient (Fig. 4b, Supplementary Fig. 4A). However, plants expressing the S299A mutant retained the response to the light gradient under high light while the S299D mutant behaved similarly than the wild type (Fig. 4c, Supplementary Fig. 4B, C). We then compared the phot1 and phot2 mutants in strong light given that PKS4L accumulates in phot2 but not phot1 (Fig. 3c). Interestingly, the phot1 mutant had a stronger gradient response than phot2 (Fig. 4d). In plants expressing PKS4 the ability to respond to a gradient in strong light was therefore only present in those lacking the ability to form PKS4L (phot1 and S299A). Collectively, these experiments indicate that in bilateral irradiation experiment, at high fluence rates phot1 prevents a response to such gradients by phosphorylating PKS4 (Fig. 4, Supplementary Fig. 4).

## Discussion

To study the function of the PKS4L isoform we identified a serine residue (S299) that is phosphorylated in response to light and is essential for the light-induced mobility shift due to PKS4 phosphorylation[19] (Fig. 1). Consistent with being phosphorylated by phot1 S299 is part of a PKA phosphorylation consensus sequence[27] (Fig. 1a, Supplementary Fig. 1E). Our data indicate that the light-induced PKS4 mobility on SDS-PAGE gels either result from phosphorylation of S299 itself or results in ordered phosphorylation events with S299 phosphorylation being the initial trigger[31]. Substituting S299 either into A or D did not alter PKS4 association with the plasma membrane or interaction with phot1 and NPH3 (Supplementary Fig. 1). Moreover, plants expressing those variants maintained rapid phot1-mediated NPH3 dephosphorylation and the phototropic response to low blue light fluence rates (Supplementary Fig. 1F, 2A). We therefore conclude that PKS4 including the S299A and S299D variants are functional for phototropism at least in some conditions. Given that S299A rescues the phototropic defect of pks4, we conclude that the phosphorylated PKS4L isoform is not required to

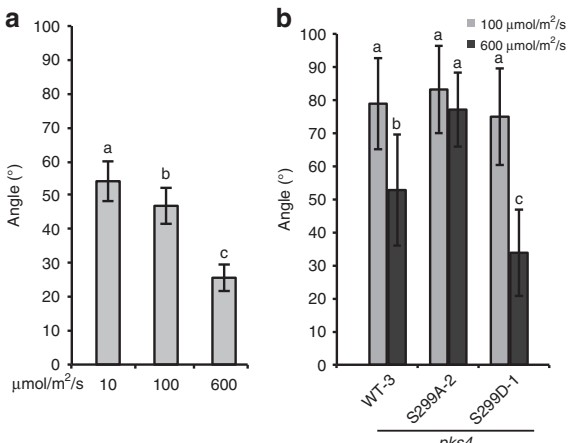

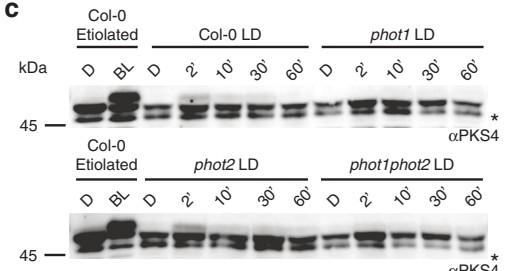

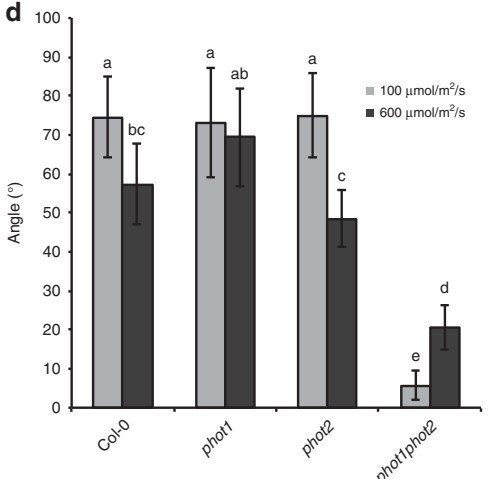

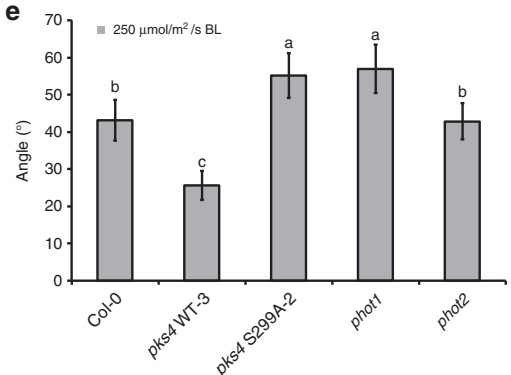

**Fig. 3** High light inhibits phototropism in a phot1- and S299-dependent manner. **a** Phototropic curvature of Col-0 seedlings treated with 10, 100, or 600 µmol m$^{-2}$ s$^{-1}$ unilateral blue light for 3 h prior to measurements. n = 68−97. **b** Phototropic curvature of pks4 PKS4-WT-3, pks4 S299A-2, and pks4 S299D-1 seedlings treated with 100 or 600 µmol m$^{-2}$ s$^{-1}$ unilateral blue light for 6 h prior to measurements. n = 25−43. **c** Western blot probed with anti-PKS4 antibodies from long-day (LD) grown Col-0, phot1, phot2, and phot1phot2 samples exposed to 600 µmol m$^{-2}$ s$^{-1}$ blue light for 0, 2, 10, 30, or 60 min. Samples from etiolated Col-0 seedlings are used as a control. Asterisks indicate unspecific band. **d** Phototropic curvature of Col-0, phot1, phot2, and phot1phot2 seedlings treated with 100 or 600 µmol m$^{-2}$ s$^{-1}$ unilateral blue light for 6 h prior to measurements. n = 29−69. **e** Phototropic curvature of seedlings of Col-0, pks4 PKS4-WT-3, pks4 S299A-2, phot1, and phot2 placed in a greenhouse at ZT4, blue light was ~250 µmol m$^{-2}$ s$^{-1}$. Phototropic bending was measured after 1 h and 45 min. n = 109−154. Seedlings were grown for 3 days in a 16−8 h day−night cycle, the experiment started 2 h after dawn (ZT2) except for panel (**e**). Data represent means. Error bars correspond to 2× standard error. Means with same letter are not significantly different (p > 0.05, one-way ANOVA test for (**a**) and (**e**), two-way ANOVA for (**b**) and (**d**))

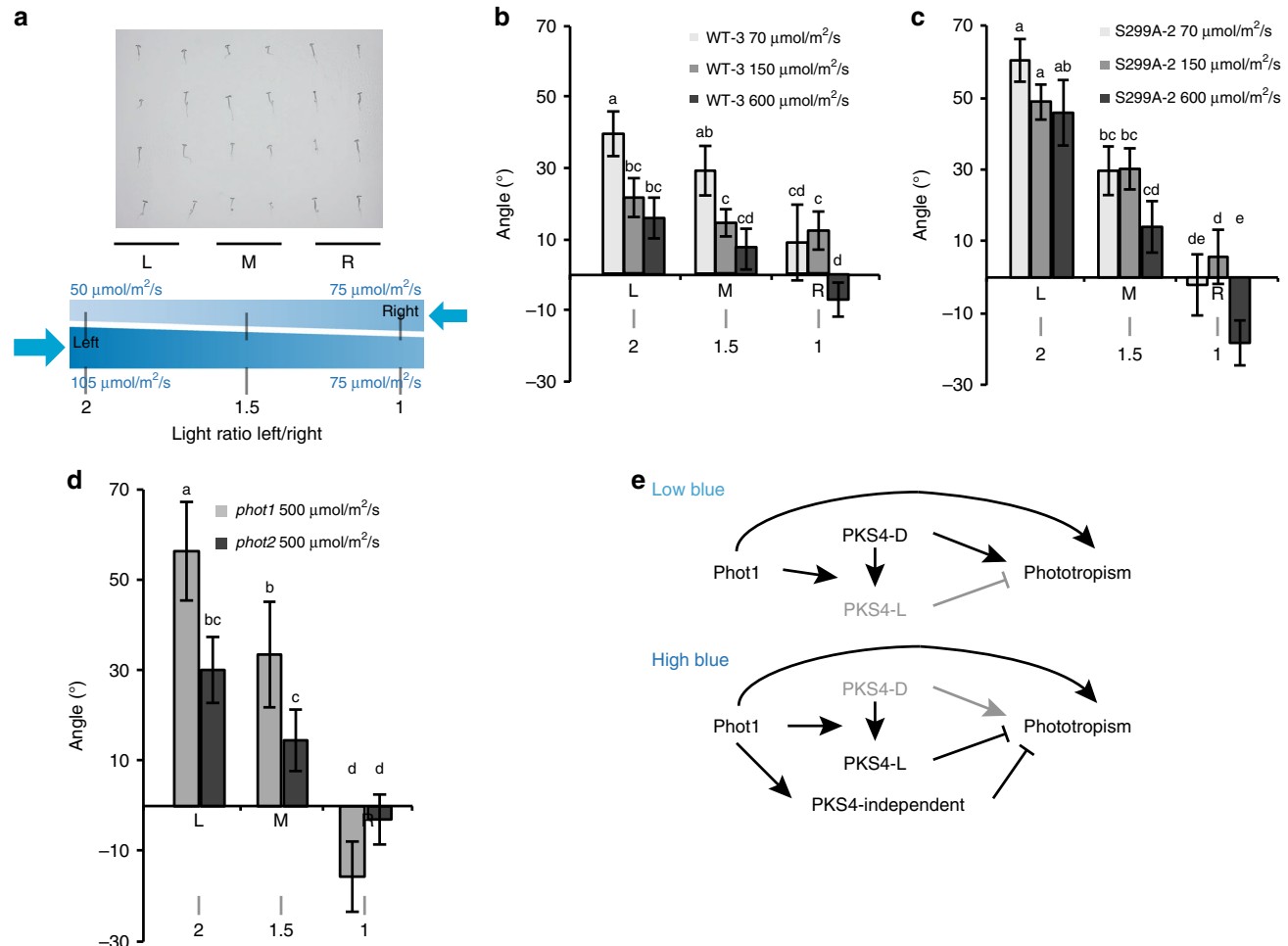

**Fig. 4** phot1 and S299 are needed for a normal response to light gradients. **a** Graphic representation of the experimental setup. Blue numbers on top represent blue light intensity coming from the right on the left and right side of the plate. Blue numbers on the bottom represent blue light intensity coming from the left on the left and right side of the plate. Black numbers represent the light ratio between the left and right sides. L, M, and R represent two columns of seedlings on the Left, Middle, and Right side of the plate, respectively. Note that seedlings in the L position are exposed to approximately a 2/1 light ratio while those in the R position get about the same intensity from each side. **b** Phototropic curvature of *pks4* PKS4-WT-3 seedlings treated with 400/105/50 μmol m$^{-2}$ s$^{-1}$ from the left and 300/75/30 μmol m$^{-2}$ s$^{-1}$ from the right. These values maintain the same gradient across the plate at a total light intensity of 600/150/70 μmol m$^{-2}$ s$^{-1}$. $n = 82-91$. **c** Phototropic curvature of *pks4* S299A-2 seedlings treated as in panel (**b**). $n = 82-91$. **d** Phototropic curvature of *phot1* and *phot2* seedlings treated with 360 μmol m$^{-2}$ s$^{-1}$ from the left and 240 μmol m$^{-2}$ s$^{-1}$ from the right (500 μmol m$^{-2}$ s$^{-1}$ total light), maintaining the same gradient across the plate as in (**b**, **c**) at a high overall intensity. $n = 15-26$. Seedlings were grown as in Fig. 3. Data represent means of phototropic bending measured after 7 h of light treatment. Error bars represent 2× standard error. Means with same letter are not significantly different ($p > 0.05$, two-way ANOVA test). Positive values correspond to bending towards the left of the plate, negative values correspond to bending towards the right of the plate. **e** Simplified model for the role of PKS4 which is enhancing phototropism in low blue light, while inhibiting phototropism in high blue light following phot1-mediated phosphorylation. The inhibitory action of PKS4-L could be through inhibition of phot2-mediated phototropism, a possibility that we have not explicitly included in the model for simplicity

promote phototropism. In this sense, phot1-mediated PKS4 phosphorylation differs from BLUS1 phosphorylation, which is an essential activating step in the latter[23,24]. Phosphorylation-mediated inhibition or activation depending on the substrate has also been observed for other protein kinases as exemplified in this study of a cyclin-dependent kinase[32].

Based on evolutionary arguments it was proposed that it is easier to evolve an inhibitory mechanism through protein phosphorylation from an active ancestor because it is more difficult to envisage the trajectory from an initially inactive protein needing to evolve the ability to be phosphorylated for activation[33]. We propose a variation of such an evolutionary scenario with PKS4D acting as an enhancer of phototropism while phot1-mediated phosphorylation leads to the formation of a PKS4 isoform inhibiting phototropism. Etiolated seedlings need to

establish photo-autotrophy and are exquisitely sensitive to light gradients[11,30]. PKS4 is specifically required for phototropism in response to low light[29] (Supplementary Fig. 2G). On the other hand we show that in etiolated seedlings phot1-mediated formation of PKS4L[19] delays and inhibits phototropic responses to higher light (Fig. 2, Supplementary Fig. 2). The phenotype of plants expressing S299D is consistent with the hypothesis that S299D partially mimics PKS4L in etiolated seedlings (Fig. 1, Supplementary Fig. 2F). Similarly, in green seedlings we present evidence for a phototropism inhibitory function of the S299D mutant (Fig. 3, Supplementary Figs. 3, 4). However, plants expressing this PKS4 variant still display responses to light intensity indicating that this variant is not fully locked in one state (Fig. 3b, Supplementary Fig. 3B). It is not uncommon that phosphomimic substitutions do not fully mimic phosphorylation,

which can be explained based on the chemical difference between a phospho-S (2 negative charges) and a D (1 negative charge)[34]. We propose that the right balance of PKS4D and PKS4L provides Arabidopsis with the ability to respond to very low light while at the same time prevent an excessive response when there is plenty of light.

In green seedlings the phototropic response is inhibited in favorable light conditions[2,3,10]. However, whether this directly depends on phototropin signaling remains unclear. Here we show that phot1 and PKS4L limit phototropism in high light. Plants expressing PKS4L but not phot1 and S299A mutants (neither of which accumulate PKS4L) limited phototropism in high light (Fig. 3, Supplementary Fig. 3). Moreover, in realistic light conditions (greenhouse) phot1 and S299A expressing plants showed a stronger phototropic response than the wild type or phot2 (Fig. 3, Supplementary Fig. 3). To study phototropism in more shallow light gradients we used bilateral illumination, a protocol used before to show that the phototropic response integrates information from multiple light sources[35,36]. Our data using light-grown Arabidopsis seedlings are consistent with previous findings when the overall light intensity is low (Fig. 4). Moreover, we show that increasing light intensity reduces seedling sensitivity to light gradients, a process also regulated by phot1 and PKS4 phosphorylation (Fig. 4, Supplementary Fig. 4). A role for phot1 in limiting phototropism of etiolated seedlings in strong light was reported in some but not all previously studies[7,37]. Another response promoted by phot1 in low light but inhibited in high light is chloroplast accumulation[38]. Similarly, phot2 also promotes chloroplast accumulation towards low blue light and mediates the chloroplast avoidance response towards high light[39]. In green seedlings we used several assays, which consistently show that phot1 limits phototropism in high light (Figs. 3 and 4). We uncover PKS4 phosphorylation as one mechanism contributing to this response. However, our data also show that phot1 also limits phototropism in high light through additional mechanisms (Supplementary Fig. 3A, C). For example, RPT2 has also been implicated in de-sensitizing phototropism in etiolated seedlings[22]. We propose that in pks4 other phot1-mediated mechanisms preventing phototropism are deployed. This may involve other members of the PKS family, which contribute to the regulation of phototropism[17,29,37]. Although such mechanisms remain to be discovered, our study identifies a phosphorylation switch that limits the phototropic response in strong light (Fig. 4e). Future studies are required to reveal the physiological importance of inhibiting phototropism in high light environments.

Based on secondary structure predictions PKS4 is a largely intrinsically disordered protein with rather low sequence conservation among orthologs. In this context it is noteworthy that S299 is a conserved residue among angiosperms indicating possible conservation of the regulatory mechanism uncovered in Arabidopsis (Fig. 1a). Indeed, a recent comparative phospho-proteomics study across 18 yeast species revealed that conserved ancient phosphorylation motifs are more likely to be functionally important[40]. An important question that remains to be solved is the biochemical consequence of PKS4 phosphorylation. This is both a fascinating and difficult question as intrinsically disordered proteins have emerged as important signaling elements[41] and because such proteins are particularly challenging to characterize biochemically.

## Methods

**Plant material and growth.** All plants used in this study are in the *Arabidopsis thaliana* Columbia-0 (Col-0) background. The mutants used were *pks4*-2, *phot1*-5, *phot2*-1, and *phot1*-5 *phot2*-1[29]. HA-epitope-tagged PKS4 wild-type and different PKS4 mutants were obtained using standard cloning methods (described below).

These PKS4 variants were placed under the control of the PKS4 promoter and the constructs were transformed into *pks4*-2. Several single insertion lines were characterized for each construct. PKS4 variants in the *pks4*-2*phot1*-5 background were obtained by crossing and verified by genotyping.

For seed production, plants were grown on soil at 22 °C with 16 h light per day (PAR ~250 µmol m$^{-2}$ s$^{-1}$). For physiological experiments, seeds were surface-sterilized in 70% ethanol and 0.05% Triton-X for 3 min and in 100% ethanol for 3 min. Seeds were sown on Petri dishes containing half-strength Murashige and Skoog medium, 0.7% phytagar. Plates were stored in the dark for 3 days at 4 °C, germination was induced by 3–6 h of white light (80 µmol m$^{-2}$ s$^{-1}$) at 22 °C. For dark-grown seedlings, plates were placed vertically in darkness at 22 °C after induction of germination. For light-grown seedlings, plates were placed vertically in long day conditions (16 h ~150 µmol m$^{-2}$ s$^{-1}$ white light from fluorescent lamps/8 h dark).

**Generation of transgenic lines.** A sequence containing 1.5 Kb PKS4 promoter, PKS4 CDS, and a *Bam*HI site right after the last PKS4 codon (no STOP) was amplified from genomic DNA and cloned as a *Hin*dIII-*Bam*H1 fragment into pBSSK (pPS1). A PKS4 sequence with S80, S85, S86, S122, S129, S130, T258, S259, and S299 mutated into alanine was ordered from eurofins®. This plasmid was digested with *Nru*I and *Bam*HI. A 1193 bp fragment of PKS4 containing all the mutated sites was then used to replace that sequence in pPS1 to yield pBSSK containing PKS4::PKS4 with nine sites mutated (pPS3). A plasmid containing only the S299A mutation was obtained by replacing the *Nco*1-*Bam*H1 fragment of pPS1 with *Nco*1-*Bam*H1 fragment of pPS3 (called pPS7). The same strategy was used to obtain the S299D mutant. We ordered a plasmid from eurofins® with the same residues mutated into aspartate. The S299D construct in pBSSK was called pPS20. These plasmids were sequenced to check that no extra mutations were inserted. pPS1, pPS3, pPS7, and pPS20 were then digested with *Bam*HI and *Hin*dIII. The 2732 bp fragment containing PKS4pro::PKS4 were extracted and ligated into pCF398 previously digested with *Bam*HI and *Hin*dIII. pCF398 is a binary plant transformation vector containing a CaMV 35S promoter, a 3xHA tag for C-terminal fusions, and the GFP gene driven by the seed-specific promoter At2S3 for selection of transgenic plants. pCF398 was constructed by inserting a triple HA cassette into pFP101[42]. In the resulting constructs the CaMV 35S promoter is replaced by PKS4pro::PKS4 sequences resulting in WT, "all-A", S299A and S299D with C-terminal 3XHA respectively called pPS9, pPS10, pPS14, and pPS22. A similar strategy was used for the GFP-tagged constructs, starting from a vector containing PKS4pro::PKS4::GFP (pED10), the S299A and S299D variants were called pPS27 and pPS28 respectively. Transgenic lines were obtained using *Agrobacterium tumefaciens*-mediated transformation of *pks4*-2[43]. For each construct we selected several single insertion lines.

**Phototropism measurements.** For etiolated seedlings we used previously described methods[29,44]. Briefly, seedlings were grown on vertically orientated plates for 3 days in darkness at 22 °C prior to the light treatments. Plates were pictured using an infrared camera. For time courses, pictures were taken every 15 or 30 min using an infrared CCD camera system[44]. For time course experiments only seedlings with cotyledons on the opposite side of the incident light source were selected for measurement as the speed of phototropic re-orientation is influenced by the position of the cotyledons[29]. Angles formed by the hypocotyls relative to vertical were measured with the ImageJ software. For green seedlings, plants were grown for 3 days in 16/8 photoperiods, phototropic experiments were typically done at ZT2 on day 4. Phototropic bending angles were calculated by subtracting average angle of orientation of upper region (85−95% of total length) of each hypocotyl with respect to horizontal after blue light treatment determined by a customized MATLAB script developed in the Fankhauser lab. For phototropic experiments in greenhouse, seedlings were placed in a greenhouse from the University of Lausanne in boxes facing the south.

**Protein analysis.** Total protein extracts and microsome factions, separation on SDS-PAGE gels and western blotting were performed as in refs. [17,19]. Briefly, total proteins (100 µl 2× Laemmli buffer for 20 mg fresh weight; 10 µg per lane) were separated on 8% SDS/PAGE gels and transferred onto nitrocellulose with 100 mM CAPS pH11 + 10% (v/v) ethanol. Original blot images for Figs. 1 and 3 can be found in Supplementary Figs. 5 and 6. For immunoprecipitation, we used the µMACS system (Miltenyi Biotec). The microsome fractions were incubated with 40 µl of µMACS GFP-tagged magnetic MicroBeads for 1 h on ice and then loaded on µcolumns, mounted on the µMACS separator, prepared with 200 µl extraction buffer. After four rounds of beads washing with extraction buffer, the eluates were extracted with 70 µl of 2× Laemmli buffer at 95 °C. Proteins were separated on 8% acrylamide SDS-PAGE gels. Anti-PKS4 antibodies were used at 1/500[19], anti-HA at 1/2000 (3F10 monoclonal, Roche) anti-phot1 at 1/5000[17], anti-NPH3 at 1/3000[17] and anti-DET3 at 1/20,000[17] dilutions in 1× PBS containing 0.1% Tween-20 and 5% non-fat milk. Chemiluminescence signals were generated using Immobilon Western HRP Substrate (Millipore). Signals were detected with a Fujifilm ImageQuant LAS 4000 mini CCD camera system and quantifications were performed with ImageQuant TL software (GE Healthcare).

**Cantharidin treatment**. Nylon meshes with 3-day-old etiolated seedlings were transferred onto freshly prepared plates supplemented by 10 μM cantharidin (Sigma) in 1% DMSO, 2 h before the light treatments.

**Microscopy**. Seedlings were imaged using an inverted Zeiss confocal microscope (LSM 710 INVERTED, ×10 objective). GFP excitation was obtained using an Argon laser at 488 nm. Pinhole was opened at 51 μm. Images were processed with the ImageJ software.

**Identification of putative phosphorylation sites**. PKS4 orthologs were identified using the OMA browser[25]. To identify putative PKA phosphorylation sites we used PhosphAT[26]. These analyses yielded the following seven residues as potential candidates: S80, S85, S86, S122, S129, S130, and S299. A meta-analysis of protein mass spectrometry identified T258 and S259 as repeatedly phosphorylated residues[28].

**Mass spectrometry**. We immunoprecipitated PKS4-GFP from etiolated seedlings (dark) or etiolated seedlings treated for 1 min with strong blue light to promote PKS4L formation (light). Protein extracts were run on SDS-PAGE, the gel area cut out around the size of the protein digested with chymotrypsin (Promega) as described[45]. Extracted tryptic peptides were dried and resuspended in 0.05 % trifluoroacetic acid, 2% (v/v) acetonitrile for mass spectrometry analyses. Peptide mixtures were injected on an Ultimate RSLC 3000 nanoHPLC system (Dionex, Sunnyvale, CA, USA) interfaced to a high-resolution mass spectrometer QExactive Plus (Thermo Fisher, Bremen, Germany). Peptides were loaded onto a trapping microcolumn Acclaim PepMap100 C18 (20 mm × 100 μm ID, 5 μm, Dionex) before elution on an Easy Spray C18 PepMap column (50 cm × 75 μm ID, 2 μm, 100 Å, Dionex) at a flowrate of 0.25 μl per min. A gradient from 4 to 76% acetonitrile in 0.1% formic acid was used for peptide separation (total time: 140 min). Full MS survey scans were performed at 70,000 resolution. In data-dependent acquisition controlled by Xcalibur 4.0 software, the ten most intense multiple-charged precursor ions detected in the full MS survey scan were selected for higher energy collision-induced dissociation (HCD, normalized collision energy NCE = 27%) and analysis in the orbitrap at 17,500 resolution. An isolation window of 1.5 m/z units around the precursor was used and selected ions were then dynamically excluded from further analysis during 15 s.

MS data were analyzed using Mascot 2.6 (Matrix Science, London, UK) set up to search the SwissProt and Trembl (www.uniprot.org) databases restricted to A. thaliana (November 2016 version) and including common contaminants (keratins, digestion enzymes, etc.) (77,084 sequences). Chymotrypsin (semi-specific cleavage after F, L, W, Y, excepted before P) was used as the enzyme definition, allowing two missed cleavages. Mascot was searched with a parent ion tolerance of 10 ppm and a fragment ion mass tolerance of 0.02 Da. Iodoacetamide derivative of cysteine was specified in Mascot as a fixed modification. N-terminal acetylation of protein, oxidation of methionine, phosphorylation of serine, threonine and tyrosine were specified as variable modifications.

Scaffold software (version 4.8.4, Proteome Software Inc., Portland, OR) was used to validate MS/MS-based peptide and protein identifications, and to perform dataset alignment. Peptide identifications were accepted if they could be established at greater than 80.0% probability by the Scaffold Local FDR algorithm. Protein identifications were accepted if they could be established at greater than 95.0% probability and contained at least five identified peptides. Protein probabilities were assigned by the Protein Prophet algorithm[46]. Proteins that contained similar peptides and could not be differentiated based on MS/MS analysis alone were grouped to satisfy the principles of parsimony. Proteins sharing significant peptide evidence were grouped into clusters. MsViz[47] software was used to compare sequence coverage and phosphorylation of the PKS4 protein between the dark and light treatments. We obtained a coverage of 59 and 54% of the dark- and light-treated samples respectively. We identified the following phosphorylated residues: S89, S183 or S184 in both samples, S200 and S299 only in the light-treated sample.

**Statistical analysis**. We performed one-way ANOVA (aov) and Compute Tukey's Honest Significance Differences (HSD.test) [agricolae package] using the R software. We considered $p < 0.05$.

**Data availability**. The authors declare that all (other) data supporting the findings of this study are available within the manuscript and its supplementary files or are available from the corresponding author upon request.

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

## Acknowledgements

This work was supported by the University of Lausanne, a grant from the Swiss National Science Foundation (3100A-160326) to C.F. We thank the Cellular Imaging Facility (University of Lausanne) for help with microscopy.

## Author contributions

Conceptualization, P.S., E.D., and C.F.; Investigation, P.S., E.D., P.W., L.A.P., and M.T.; Writing, C.F. with contributions from P.S. and E.D.; Supervision, C.F.

## Additional information

**Competing interests:** The authors declare no competing interests.

