## [Peer Review File · Nature Communications]

Reviewers' comments:

Reviewer #1 (Remarks to the Author):

I enjoyed reading this work, which advances our understanding of phototropism in complex lighting regimes more representative of real world conditions. The authors assess the functional role of a conserved residue (S299) within the PKS4 protein, which modulates the phototropic response to blue light. Interestingly, the authors demonstrate a fluence rate-dependent switch that modulates plants' phototropic response.

The methods are soundly described, and statistical tests are appropriate.

My main difficulty surrounds the phenotype of the PKS4-S299D line. This protein is biologically active (as it delays phototropism under low light; Figure S2F, and has reduced curvature under 600uE blue light; Figure 3B). However, the S299D transgene has no effect in response to light gradients (Figure 4), nor under higher light conditions (Figure S3). Why? If S299D mimics PKS4L (via a phosphomimic) then why do these plants not suppress phototropism? The authors address this partially in their discussion, but by what mechanism would S299D only be functional in certain biological contexts? It feels a little unsatisfying to have this loose end, and adds to the uncertainty regarding the underlying mechanism, especially given the authors' inability to identify S299 as a target of phosphorylation *in vivo*.

The authors use transgenic lines expressing WT PKS4 as a control. I'd prefer to see *pks4* mutants presented throughout as an additional control, given that this is the genetic background of the transgenic lines created. This is particularly important when interpreting the PKS4-S299D datasets, and in Figure 3C, where Col-0 and WT-2 are significantly different.

On a related note, what is the phenotype of S299D plants when treated with CN? If the conformational changes induced by S299D induce or reveal consequential phosphorylation events a response to CN treatment in these lines may be informative.

Figures

Figure 1- Why are 2 residues highlighted in orange? Only 1 is described in the legend.

Figure 3- In Figure 3B, the curvature of S299D plants is more pronounced than WT at 600uE. But what is observed at 10uE?

Figure 3- I would have liked to have seen S299D included in Figure 3E.

Minor comments

1) I would add an indication to the figures that PKS4-HA constructs were introduced into a *pks4* background- this isn't currently clear, and was only explicit when reading the manuscript text.

2) "exemplified by the big difference in PKS4L formation between etiolated seedlings treated with 0.1 and 10 $\mu\text{mol m}^{-2} \text{s}^{-1}$ blue light (figure S2C)"

Are the authors referring to the proportion of PKS4L relative to PKS4D? If the authors are comparing 0.1 vs 10 uE blots additional controls will be needed (e.g. correction for exposure time, loading controls etc)

3) "Moreover, the *pks4* phenotype indicates that other processes also modulate this response." Could the authors please clarify which *pks4* phenotype they are referring to?

4) "phosphorylated PKS4L isoform is not required to activate the protein."

Could the authors please clarify- which protein is activated? PKS4? If this is the case, what do the authors mean by 'activated'?

5) Supplemental Figure 2- The four tones of gray used in Figure S2D are hard to distinguish; could the authors use color instead?

6) Supplemental Figure 3
Missing y axis (should be Angle?)

7) uE are used in Figure 2 panel A, but uMolm⁻²s⁻¹ is used elsewhere (also in Figure S4).

Reviewer #2 (Remarks to the Author):

The ms by Schumacher et al. described an experiment to investigate the functional significance of PKS4 phosphorylation by Phot1. They first analyzed phosphosites of PKS4 and determined S299 being the critical phosphosites. They then used transgenic plants expressing S299A and S299D to investigate the role of phosphorylation of this site residue of PKS4. The authors discovered that phot 1-dependent phosphorylation of PKS4 is required for inhibition of phototropism in high light. Although it is intuitive that plants need to not only sense light but also limit their light response, exactly how such systems work is extremely difficult to solve. The authors of this ms used a unique yet specific approach to convincingly demonstrate that phot 1 phosphorylation of S299 is part of such a mechanism in phototropism. It is particularly interesting that the authors designed several remarkably elegant experiments to address the high light and light gradient problems as plants would normally experience in nature. The paper is well written and ready to be published in NC with some minor clarifications.

1. The authors described that "We used a combination of MS and" to come up with the 9 putative phosphosites for study. However, according to what described in the Method, it seems that 4 low confidence phosphosites of PKS4 were detected by MS, none of them are light-responsive, whereas other 5 phosphosites including S299 are completely computational predicted. It seems much better to this reviewer if the authors may avoid the vaguer and confusing description of "a combination of" Instead, it would make more sense to simply describe the whole process in the main text, including the failure of the MS analysis and the success of the computational analysis, and by clarifying the fact that S299 was identified completely based on the computational prediction.

2. Fig. 1, it is not clear why R is also labelled with orange. Moreover, the authors may want to discuss possible implications that PKS4 from two plants, *Theobroma* and *Musa*, contain no S299 equivalent residue.

3. Fig. 2B, it may make it clearer if the 10 micromole curve from A is duplicated here as a dashed curve for the control. Moreover, in this figure and other figures, the authors used WT2, WT3, etc, but it seems not clear from legends what are those.

4. Fig. 2C, what are the level of significance or the p values?

5. Fig. 3C, what is LD? It seems not explained in legend

6. Fig. 4, there is a large empty space in this figure that seems perfect to add a working model to explain the authors' overall discovery and hypothesis, which would likely make it easier for readers to better understand this complex story.

Reviewer #3 (Remarks to the Author):

In this report, Schumacher and colleagues address the role of PKS4 in phototropin photoreceptor mediated phototropic signaling. PKS1 (Phytochrome Kinase Substrate 1) was first identified as interactor and substrate of phytochrome red/far-red photoreceptors and later found to interact with phototropin (phot) blue-light receptors. Previously, the corresponding author's group has shown that PKS4 was phosphorylated by phot1 and not by phot2. Here, the authors report the role of phospho S299 residue of PKS4 protein in detail. Mutating Ser299 into Ala abolished the phosphorylated form of PKS4 seen in light (PKS4-L) and the phosphomimetic S299D gave it slower mobility indicating constitutive phosphorylation. It is concluded that PKS4-L slows down phototropism.

Overall, this manuscript is poorly written, and some parts are not scientifically concrete. In its current state, this paper needs substantial improvement before it can be published. The data on the role of PKS4-L form in inhibiting phototropism at high-light intensity is not conclusive and have not ruled out other alternatives. The data in this manuscript is raising more questions than it is attempting to answer. At present, this manuscript will not add more and move the field forward from what is already known on PKS4 from the previous work of Demarsy et al. (2012, EMBO J) published from the corresponding author's lab. My comments to improve the manuscript are listed below:

1. The explanation of the nature of PKS4 band in the S299D (phosphomimetic) transgenic line is not satisfactory (page 5 and figure 1C). Does the slowly migrating form of PKS4 look similar to the WT version after treating it with a phosphatase? Does the S299D mutant lead to constitutive phosphorylation of PKS4?
2. I could not find a single immunoblot presented with loading controls. The methods section does not indicate how proteins were prepared and ensured that equal quantity of the sample was loaded into each lane before the immunoblot analysis. This practice is not scientifically appropriate and does not give the impression that the experiments were robust enough.
3. Page 4: "...with NPH3 release from the plasma membrane and phot1 signaling de-sensitization..." is factually wrong.
4. Page 5: "We used a combination of immunoprecipitation followed by mass spectrometry and phosphosite predictions considering that phot1 is an AGC kinase and site conservation among angiosperms PKS4 orthologs (Figure 1A) (Materials and methods)". It is indicated that the method for phosphosite prediction is described in the methods. However, I could not locate it.
5. Figure 2A-B, S2D-E: The authors show that S299A transgenic lines do not accumulate PKS4-L form showed lower phototropic responsiveness at 10 micromol blue-light compared to the PKS4-WT lines. However, it was difficult to conclude with the inconsistent usage of lines between different experiments. PKS4 WT-2 line is used in Figure 2A whereas WT-1 and WT-3 lines are used in Figure S2D. What is the phototropic curvature of all these lines? This gives an impression that cherry-picking of data was employed and it will be hard to draw conclusion. Also, I am surprised that S299D phosphomimetic line has not been used in this assay, which will be useful in supporting the conclusions. What is the phenotype of S299D at 10 micromols? Figure 2A and B can be combined to easily grasp the data.
6. Figure 2C and other figures: As detailed above, what is the phototropic curvature for multiple lines of PKS4 WT, S229A and S229D? Consistent usage of the same lines will be helpful to interpret the data.
7. Figure 3A-B: How does the PKS4 protein behave at 10, 100 and 600 umol of blue-light? What is the phenotype of S299A in phot1 mutant? This will be important to test the author's hypothesis that PKS4-L helps phot1 to inhibit phototropism at high-light intensity.
8. The discussion is mostly redundant with the results. The authors do not really discuss how PKS4 and phot1 could prevent phototropism in high-light.

9. Figure S1D: It appears that immunoblot of "plasma membrane GFP" and "WT-GFP" are spliced together. In this case, it will be appropriate to indicate the splice with a line.
10. Figure 1A: As per the nomenclature, the genus is abbreviated and not the species. It's *A. thaliana* and not "Arabidopsis t."
11. This sentence on page 8 is not clear. "In contrast, depending on the position of the sun and the plant in its community green seedlings experience moments with strong light gradients."

Reviewer #4 (Remarks to the Author):

The work by Schumacher et al. aims to elucidate the functional role of the phosphorylated form of Phytochrome Kinase Substrate 4 (PKS4) in plants. Authors previously found that the phosphorylated form of PKS4 (PKS4L) was produced in response to blue light, and the phosphorylation is induced by phototropin kinase. PKS4L was identified as the low mobility form of PKS4 on SDS-PAGE. In this work, authors conclude that phot1 has the dual roles: stimulation of phototropism under the low light and inhibition of phototropism under the high light through PKS4L. The results are clear and interesting, but I am not confident of the novelty of this work.

Conclusion of this work was drawn from the following results.

Authors first identified the phosphorylation sites in PKS4L as S299 by the combinational usage of immunoprecipitation and mass spectrometry. They also showed that the S299 phosphorylation caused the low mobility and phosphomimic S299D mimicked this mobility shift (Fig. 1).

To know the functional role of PKS4L, authors initially indicated that PKS4-S299A (non-phosphorylatable form) and PKS4-S299D (phosphomimic form) revealed the similar properties with respect to localization in the cells and the interaction with phot1 and NPH3. They also showed that the mutations of S299 in PKS4 did not affect the phototropin-mediated NPH3 dephosphorylation (SFig. 1).

To elucidate the functional role of PKS4L, they generated the HA-tagged PKS4 lines by introducing the PKS4 variant genes in *pk4* mutant, and compared the kinetics of phototropism in response to low and high fluence rates of blue light. They found that WT and the PKS4-S299A mutant exhibited the similar phototropism under the low light ($0.1 \mu\text{mol m}^{-2} \text{s}^{-1}$), but the phenotype had the slower bending in WT than the PKS4-S299A mutant in response to strong blue light ($10 \mu\text{mol m}^{-2} \text{s}^{-1}$) (Fig. 2). Phosphomimic line of PKS4-S299D revealed a slower phototropic response even under the low light. Slow down of phototropic bending in WT was not found in PKS4-S299A expressed lines (Fig. 3). From these results, authors concluded that PKS4L acted as a negative regulator for the phototropic bending under the strong light.

They further provided the evidence for the slow-down of phototropic bending in WT green seedlings under the high light, and suggested that the slow-down bending was caused by the accumulation of PKS4L. However, they noted that such reduction of phototropism was also found in the mutant *pk4* that lack PKS4, suggesting other processes also work in the reduction (Fig.3, SFig.3).

Furthermore, they determined which phototropin (s) induced the phosphorylation of PKS4 in green seedlings, and showed that phot1 mediated this reaction. They provided the data that the phot1 plants exhibited a larger bending than WT and phot2 mutant in response to the strong blue light, suggesting that phot1 inhibited phototropism through PKS4 phosphorylation (Fig. 3). Since phot1 mediates the phototropism under the low light, they formulated the notion that phot1 has dual roles in both stimulation and inhibition of phototropism under the low light and the strong blue light,

respectively, through the function of PKS4.

Finally, authors illuminated green seedlings with two blue light sources from opposite sides, causing the light gradient on plants (Fig. 4, SFig.4) because such situation occurs in the natural environment. In response to the light gradient, the plants expressing S299A respond by bending, but WT and the S299D mutant did respond in the lesser extent. Furthermore, the phot1 mutant exhibited a stronger gradient response than the phot2 mutant. From the results, authors concluded that phot1 prevented phototropism under the presence of light gradient by producing PKSL4.

The experiments were carefully done and the data presented here support their conclusion.

Comments

1) I have a question on the physiological significance of the response presented here. It is unclear whether the lack of PKS4L causes adverse effects or not on plants. For example, the loss of PKS4L may cause the reduction of plant growth under the strong light. If author could provide the related data, this work becomes much better.

2) One of the key points of this work is the data for the dual role of phot1 in phototropic bending. However, such notion for phototropin is not new because phot2 also has a dual role for chloroplast accumulation and avoidance. Please note this in the text (See Kagawa, T. and Wada, M. Plant Cell Physiol. 41: 84-93, 2000).

3) It is unclear the quantitative relationship between the PKS4L production and inhibition of phototropic bending. Relationship between the light intensity, amount of the PKS4L production, and the degree of bending inhibition is preferable.

4) Description of the experiments in Fig. 4 is not easy to understand. Please describe more precisely.

Response to the reviewers' comments

Reviewer 1

It feels a little unsatisfying to have this loose end, and adds to the uncertainty regarding the underlying mechanism, especially given the authors' inability to identify S299 as a target of phosphorylation in vivo.

We now provide direct evidence for light-induced S299 phosphorylation (Figure 1B). Our mass spec facility recently obtained a new instrument that is significantly more sensitive than the previous one, presumably enabling us to finally detect this modification. We thank the reviewer for insisting on this data because it now allows us to interpret the S299A mutant with much more confidence.

A Ser to Ala mutant is relatively straight forward to interpret simply because in such a mutant phosphorylation cannot happen on that particular residue without otherwise substantially modifying the protein. This is why we have focused our analysis on the S299A mutant in which the light-induced mobility shift no longer occurs and which presents phenotypes consistent with enhanced activity of PKS4. On the other hand, Ser to Asp mutants do not necessarily fully

mimic Ser phosphorylation. One simple explanation for this is that the charge difference between Ser and phospho-Ser is greater than between Ser and Asp (referenced in the manuscript). It is therefore very difficult to ascertain that an S to D mutant is a full mimic of Ser-phosphorylation. This being said we have included new data about the S299D mutant on Figures S3C and S3D, which strengthen our interpretation that S299D is at least a partial phosphomimic in different experimental conditions: in etiolated seedlings, light-grown seedlings in the lab and in the greenhouse (data presented in figures 2B, S2F, S3B, S3C and S3D).

The authors use transgenic lines expressing WT PKS4 as a control. I'd prefer to see *pks4* mutants presented throughout as an additional control, given that this is the genetic background of the transgenic lines created. This is particularly important when interpreting the PKS4-S299D datasets, and in Figure 3C, where Col-0 and WT-2 are significantly different.

We now also include the phenotype of *pks4* in new S2G and S3D. Regarding Figure 3C, this must be a mistake as Figure 3C is a western blot. I presume that the reviewer meant 3B. All the lines in figure 3B are in a *pks4* mutant background. Hence what we are comparing is the PKS4-WT, S299A and S299D transgenes (expressed at similar levels see S2B). In my opinion this is the most direct possible comparison. The comparison between Col and *pks4* is provided in figure S3A and new S3D.

On a related note, what is the phenotype of S299D plants when treated with CN? If the conformational changes induced by S299D induce or reveal consequential phosphorylation events a response to CN treatment in these lines may be informative.

As indicated above due to the inherent difficulty to ascertain that a Ser to Asp mutant behaves as a true phospho-mimic our primary focus was not the S299D mutant. This being said, the clear effect of CN on *pks4* transformed with WT PKS4 but not in the S299A transformant provides strong evidence for an inhibitory effect of S299 phosphorylation.

Figures. Figure 1- Why are 2 residues highlighted in orange? Only 1 is described in the legend.

We have highlighted the R and the S of the PKA consensus, given that phot1 the presumptive protein kinase is part of the large PKA/PKC family.

Figure 3- In Figure 3B, the curvature of S299D plants is more pronounced than WT at 600uE. But what is observed at 10uE?

Given the rather small difference observed in the wild type between 10 and 100 microE (3A), we have focused our analysis on 100 and 600 microE.

Figure 3- I would have liked to have seen S299D included in Figure 3E.

We have now also included a new figure S3D, which compares the phenotypes of Col-0, *pk4* and *pk4* transformed with either WT PKS4, S299A or S299D. This data is consistent with the inhibitory role of PKS4 phosphorylation based on both the S299A and S299D phenotypes.

Minor comments

1) I would add an indication to the figures that PKS4-HA constructs were introduced into a *pk4* background- this isn't currently clear, and was only explicit when reading the manuscript text.

Thanks for the suggestion, we have now made that clear by including this information also in Figure S2A and S2B. These figures are the first descriptions of the lines used throughout the manuscript.

2) "exemplified by the big difference in PKS4L formation between etiolated seedlings treated with 0.1 and 10 μ mol m⁻² s⁻¹ blue light (figure S2C)"
Are the authors referring to the proportion of PKS4L relative to PKS4D? If the authors are comparing 0.1 vs 10 uE blots additional controls will be needed (e.g. correction for exposure time, loading controls etc)

Figure S2C illustrates what we have previously described in detail in Demarsy et al., 2012. The best way to determine light-induced PKS4 phosphorylation is to compare the ratio of the slower and faster migrating isoforms at different time points (Importantly PKS4 protein levels do not change significantly over the first hour of light irradiation, see Demarsy et al., 2012). At 10 microE in etiolated seedlings after 30" there is more PKS4-L than PKS4-D, while at 0.1 we only see a small fraction of PKS4-L after 10 minutes. As this is fully consistent with previously published work we argue that a full repetition is not useful here. However, we can simply cite previous work and remove this panel if the reviewer finds this more appropriate.

Loading controls were used to quantify PKS4 levels in the different lines used in the study (S2B). The methods are described in the supplementary information.

3) "Moreover, the *pk4* phenotype indicates that other processes also modulate this response." Could the authors please clarify which *pk4* phenotype they are referring to?

Our genetic data comparing Col-0 with *pk4* and *phot1* with *phot1pk4*, indicate that the phosphorylation state of PKS4 alone cannot explain why *phot1* mutants are more phototropic than the wild type in strong blue light. We have changed the discussion and included a model figure (4E) to clarify this.

4) "phosphorylated PKS4L isoform is not required to activate the protein."
Could the authors please clarify- which protein is activated? PKS4? If this is the case, what do the authors mean by 'activated'?

Thanks for pointing out this ambiguity. We have changed the phrase, now "Given that S299A rescues the phototropic defect of *pk4*, we conclude that the

phosphorylated PKS4L isoform is not required to promote phototropism”.

5) Supplemental Figure 2- The four tones of gray used in Figure S2D are hard to distinguish; could the authors use color instead?

We have now included color to improve readability.

6) Supplemental Figure 3. Missing y axis (should be Angle?)

Thank you, corrected.

7) μE are used in Figure 2 panel A, but $\mu\text{Molm-2s-1}$ is used elsewhere (also in Figure S4).

Thank you for pointing this out, we have now consistently used $\mu\text{M}/\text{m}^2/\text{s}$

Reviewer #2 (Remarks to the Author):

1. The authors described that “We used a combination of MS and” to come up with the 9 putative phosphosites for study. However, according to what described in the Method, it seems that 4 low confidence phosphosites of PKS4 were detected by MS, none of them are light-responsive, whereas other 5 phosphosites including S299 are completely computational predicted. It seems much better to this reviewer if the authors may avoid the vaguer and confusing description of “a combination of” Instead, it would make more sense to simply describe the whole process in the main text, including the failure of the MS analysis and the success of the computational analysis, and by clarifying the fact that S299 was identified completely based on the computational prediction.

We now provide direct evidence for light regulated S299 phosphorylation (Figure 1B), which strengthens our interpretation of the phenotype of plants expressing S299 variants and allowed us to clarify this part of the manuscript.

2. Fig. 1, it is not clear why R is also labelled with orange. Moreover, the authors may want to discuss possible implications that PKS4 from two plants, Theobrowma and Musa, contain no S299 equivalent residue.

We marked both the R and the S because (RXS) constitutes a PKA phosphorylation consensus sequence. We find it difficult to speculate on the sequences of Theobrowma and Musa PKS4 for a number of reasons. The first is that such sequences coming from genomic efforts should first be checked carefully to ensure that indeed these species do not have this PKA consensus sequence in the PKS4 gene. This being said cacao (Theobrowma cacao) grows in tropical forest understories and hence may not typically experience high light.

3. Fig. 2B, it may make it clearer if the 10 micromle curve from A is duplicated here as a dashed curve for the control. Moreover, in this figure and other figures, the authors used WT2, WT3, etc, but it seems not clear from legends what are those.

The comparison proposed by the reviewer is an interesting one, we have presented it on Figure S2E. The different lines (e.g. WT-1) refer to the lines presented in figure S2A. We now also show that the trend is the same for independent lines on S2D.

4. Fig. 2C, what are the level of significance or the p values?

We used 0.05 throughout. This is now also indicated in the figure legends.

5. Fig. 3C, what is LD? It seems not explained in legend

We have now clarified in the legend that LD corresponds to Long Days.

6. Fig. 4, there is a large empty space in this figure that seems perfect to add a working model to explain the authors' overall discovery and hypothesis, which would likely make it easier for readers to better understand this complex story.

Thank you for the suggestion. We have now included a simple model (Figure 4E) and modified the discussion. This will hopefully clarify our message.

Reviewer #3 (Remarks to the Author):

1. The explanation of the nature of PKS4 band in the S299D (phosphomimetic) transgenic line is not satisfactory (page 5 and figure 1C). Does the slowly migrating form of PKS4 look similar to the WT version after treating it with a phosphatase? Does the S299D mutant lead to constitutive phosphorylation of PKS4?

To address this comment we now also show phosphatase treatment of S299D (new figure S1E). This variant displays slower migration than the WT also after phosphatase treatment. However, we also observe that both in dark-grown and in light-grown seedlings there is a slight downshift of S299D after phosphatase treatment but no light regulation. This data is therefore consistent with constitutive phosphorylation of S299D. In contrast, in the wild type PKS4 there is a strong effect of light, which is fully consistent with our new data providing direct evidence for PKS4 phosphorylation on S299 (Figure 1B).

2. I could not find a single immunoblot presented with loading controls. The methods section does not indicate how proteins were prepared and ensured that equal quantity of the sample was loaded into each lane before the immunoblot analysis. This practice is not scientifically appropriate and does not give the impression that the experiments were robust enough.

We understand the concern of the reviewer but would like to point out that the point of all the western blots shown here is to determine the fraction of the PKS4-L and PKS4-D isoforms. As PKS4 levels do not change over the first hour of light treatment (Demarsy et al., 2012) comparing the levels of the slower and the faster migrating isoforms at each time point is an accurate way to determine the

proportion of the light-induced PKS4-L form. We therefore don't see the point of adding a blot probed for another protein for example in figures 1C and 1D. As proposed to reviewer 1 (see above), we can remove figure S2C, which simply illustrates what we have previously shown in Demarsy et al., 2012. This being said we believe that this illustration is useful for the reader.

Quantification of PKS4 protein levels is presented in supplementary figure S2B. For this experiment we used DET3 as a loading control. We refer to Demarsy et al., 2012 regarding the methodology and described the quantification in the supplementary information (under "protein analysis").

3. Page 4: "...with NPH3 release from the plasma membrane and phot1 signaling de-sensitization...." is factually wrong.

I am pasting a part of the abstract from Haga et al., 2015, which is cited in this context "Our microscopy and biochemical analyses indicated that blue light irradiation causes dephosphorylation of NONPHOTOTROPIC HYPOCOTYL3 (NPH3) proteins and mediates their release from the plasma membrane. These phenomena correlate closely with the desensitization of phot1 signaling during the transition period from first positive phototropism to second positive phototropism. ". This is what we summarized with our phrase, I therefore do not understand this comment. If the reviewer would like to suggest that we discuss other papers in this context I would be happy to follow his/her suggestions.

4. Page 5: "We used a combination of immunoprecipitation followed by mass spectrometry and phosphosite predictions considering that phot1 is an AGC kinase and site conservation among angiosperms PKS4 orthologs (Figure 1A) (Materials and methods)". It is indicated that the method for phosphosite prediction is described in the methods. However, I could not locate it.

We now provide direct evidence for light regulated PKS4 phosphorylation, which allowed us to clarify this point. The detailed description concerning mass spectrometry is provided in the supplementary materials and methods.

5. Figure 2A-B, S2D-E: The authors show that S299A transgenic lines do not accumulate PKS4-L form showed lower phototropic responsiveness at 10 micromol blue-light compared to the PKS4-WT lines. However, it was difficult to conclude with the inconsistent usage of lines between different experiments. PKS4 WT-2 line is used in Figure 2A whereas WT-1 and WT-3 lines are used in Figure S2D. What is the phototropic curvature of all these lines? This gives an impression that cherry-picking of data was employed and it will be hard to draw conclusion. Also, I am surprised that S299D phosphomimetic line has not been used in this assay, which will be useful in supporting the conclusions. What is the phenotype of S299D at 10 micromols? Figure 2A and B can be combined to easily grasp the data.

We have now included additional data to clarify this issue. First, the phototropic response of all lines is presented in Figure S2A. Figures 2A and 2B show the data for WT-2 and S299A-2. We kept this data on 2 different panels for clarity. We

also present the data for all 3 WT lines and both S299A lines in Figure S2D which confirms that S299A mutants have a faster phototropic bending response than WT at 10 microE, as expected by our model given that S299A cannot undergo light-induced S299 phosphorylation. Moreover, S299D is slower than the WT at 0.1microE.

6. Figure 2C and other figures: As detailed above, what is the phototropic curvature for multiple lines of PKS4 WT, S229A and S229D? Consistent usage of the same lines will be helpful to interpret the data.

As the different lines behaved consistently in experiments shown on Figure S2A, S2D and S3B we have not systematically shown data for all lines but taken one representative line for each. As indicated to the editor and reviewer 1 the interpretation of the S299A mutant is quite simple because in this mutant S299 cannot be phosphorylated and hence it allows the study of S299 phosphorylation in the absence of a significant change of the protein (just one S to A change). In contrast, it is essentially impossible to ascertain that an S to D mutant behaves as a full phospho-mimic mutant. This has nothing to do with the specific case of PKS4 but is a general feature of studies using such mutations of phosphorylated residues. This being said, we now also include new data on figure S3C and S3D, which is consistent with our model. For more details please check our answer to the editor and reviewer 1.

7. Figure 3A-B: How does the PKS4 protein behave at 10, 100 and 600 umol of blue-light? What is the phenotype of S299A in *phot1* mutant? This will be important to test the author's hypothesis that PKS4-L helps *phot1* to inhibit phototropism at high-light intensity.

To address this question we have crossed S299A and S299D with *phot1* in order to obtain *pks4phot1S299A* and *pks4phot1S299D*. We observed that *phot1* and *pks4phot1S299A* have a similar phenotype (Figure S3C), which is consistent with the fact that PKS4 is not phosphorylated in *phot1* (Demarsy et al., 2012; Figure 3C). However, in *pks4phot1S299D* phototropism was severely impaired showing that in this sensitized background S299D strongly interferes with phototropism (Figure S3C). This data is consistent with our model that PKS4-L inhibits phototropism.

We could only detect significant PKS4 phosphorylation in response to strong blue light in light-grown seedlings, which is why we show the data in response to 600 microE. This is consistent with what we observed in etiolated seedlings where strong light leads to more PKS4 phosphorylation than weak light (Demarsy et al., 2012).

8. The discussion is mostly redundant with the results. The authors do not really discuss how PKS4 and *phot1* could prevent phototropism in high-light.

We have modified the discussion in light of the new data we provide.

9. Figure S1D: It appears that immunoblot of "plasma membrane GFP" and "WT-

GFP" are spliced together. In this case, it will be appropriate to indicate the splice with a line.

I apologize for this oversight, this is now corrected.

10. Figure 1A: As per the nomenclature, the genus is abbreviated and not the species. It's *A. thaliana* and not "Arabidopsis t."

Thank you for pointing this out, we have corrected the figure accordingly.

11. This sentence on page 8 is not clear. "In contrast, depending on the position of the sun and the plant in its community green seedlings experience moments with strong light gradients."

Thank you for this suggestion, we have modified this phrase to improve clarity.

Reviewer #4 (Remarks to the Author):

1) I have a question on the physiological significance of the response presented here. It is unclear whether the lack of PKS4L causes adverse effects or not on plants. For example, the loss of PKS4L may cause the reduction of plant growth under the strong light. If author could provide the related data, this work becomes much better.

This is a very interesting comment that is unfortunately very hard to address experimentally. We have not observed any obvious adverse effect in plants expressing S299A (which can't form PKS4L) grown in laboratory conditions. All we can currently say is that such plants have a faster phototropic response towards strong blue light and towards sunlight (Figures 2 and 3). We believe that to identify such potential negative effects of S299A we would have to conduct field experiments, which are complex (due to the inherently variable conditions) and currently not possible for us due to legislative reasons (transgenic plants can't be grown outside in Switzerland).

2) One of the key points of this work is the data for the dual role of phot1 in phototropic bending. However, such notion for phototropin is not new because phot2 also has a dual role for chloroplast accumulation and avoidance. Please note this in the text (See Kagawa, T. and Wada, M. Plant Cell Physiol. 41: 84-93, 2000).

Thank you for this suggestion, we have now included this reference in our discussion.

3) It is unclear the quantitative relationship between the PKS4L production and inhibition of phototropic bending. Relationship between the light intensity, amount of the PKS4L production, and the degree of bending inhibition is preferable.

I apologize but I am not sure that I understand this comment. What we can say

based on our data is that plants expressing PKS4 S299A (which can't undergo light-induced S299 phosphorylation) show a different response than the wild type to higher blue light fluence rates. In the WT higher fluence rates lead to PKS4 phosphorylation and an inhibition or delay in phototropism while that was not observed in S299A plants. Our data do not allow us to draw a quantitative relationship between PKS4 phosphorylation and the phototropic response.

4) Description of the experiments in Fig. 4 is not easy to understand. Please describe more precisely.

We have tried to improve our explanation of the experimental design of the data presented in Figure 4.

REVIEWERS' COMMENTS:

Reviewer #1 (Remarks to the Author):

This is a much-improved manuscript, and the identification of S299 phosphorylation in vivo addresses many of my concerns. The retention of a light-dependent phenotype in S299D seedlings demonstrates the need for further study of this signaling pathway, but such studies are beyond the scope of this manuscript.

Minor comments

In Figure S1D what is WT-GFP? PKS4 wild type? In all of the IP figures it is not immediately obvious what the bait was...

Line 164- Which early-phot1 mediated response is being referred to? NPH3 dephosphorylation?

I don't think Figure S3E is referenced in text?

Reviewer #2 (Remarks to the Author):

The authors have satisfactorily address most of my concerns, especially the new MS data. There are a couple of points that this reviewer does not necessarily agree with the authors' arguments, but those are primarily different opinions that do not affect the quality or publishability of the manuscript.
I think this ms is ready to go.

Reviewer #3 (Remarks to the Author):

This revision has improved significantly from its original version. Additional lines of experimental data in this revised manuscript have bolstered key conclusions made by the authors. I have few minor additional suggestions which are listed below:

Figure 2A: The "WT-2" indicated in this figure and in the text (line 154) actually refers to PKS4-WT and not Col-0 ecotype. It will be better to rename them other than "WT" to inadvertently cause confusion with the Col-0 WT ecotype.

Line 170: Rephrase "...big difference in PKS4L formation" to "...large difference.."

Reviewer #4 (Remarks to the Author):

I have no big problem for this manuscript. However, physiological relevance of the response presented should be elucidated in future. Authors should mention this in the manuscript.

RESPONSE (in blue) TO REVIEWERS' COMMENTS:

Reviewer #1 (Remarks to the Author):

Minor comments

In Figure S1D what is WT-GFP? PKS4 wild type? In all of the IP figures it is not immediately obvious what the bait was...

WT-GFP is indeed PKS4-GFP, we agree that this is misleading and hence corrected the figure by replacing WT-GFP with PKS4-GFP. All IPs were made with an anti-GFP antibody this is indicated in the figure legend

Line 164- Which early-phot1 mediated response is being referred to? NPH3 dephosphorylation?

Yes, we now specified this in the text "and NPH3 dephosphorylation, an early phot1-mediated response"

I don't think Figure S3E is referenced in text?

Thanks for pointing this out, now included "(Figure S3D, experimental design shown on S3E)."

Reviewer #2 (Remarks to the Author):

I think this ms is ready to go.

No action required

Reviewer #3 (Remarks to the Author):

I have few minor additional suggestions which are listed below:

Figure 2A: The "WT-2" indicated in this figure and in the text (line 154) actually refers to PKS4-WT and not Col-0 ecotype. It will be better to rename them other than "WT" to inadvertently cause confusion with the Col-0 WT ecotype.

Thank you for pointing this out. This phrase has now been revised "We compared the kinetics of phototropism at those fluence rates and observed that a line expressing WT PKS4 (WT-2) had a slower phototropic response at 10 than at 0.1 $\mu\text{mol m}^{-2} \text{s}^{-1}$ (figure 2A)."

Line 170: Rephrase "...big difference in PKS4L formation" to "...large difference.."

corrected

Reviewer #4 (Remarks to the Author):

I have no big problem for this manuscript. However, physiological relevance of the response presented should be elucidated in future. Authors should mention this in the manuscript.

This is indeed the case. We have now added this phrase in the discussion "Future studies are required to reveal the physiological importance of inhibiting phototropism in high light environments."